# Reinforcement Learning-Based Approach for Minimizing Energy Loss of Driving Platoon Decisions [note 1]

**DOI:** 10.3390/s23084176

**Published:** 2023-04-21

**Authors:** Zhiru Gu, Zhongwei Liu, Qi Wang, Qiyun Mao, Zhikang Shuai, Ziji Ma

**Affiliations:** 1College of Railway Transportation, Hunan University of Technology, Zhuzhou 412007, China; guzhiru@hut.edu.cn (Z.G.); 16401700232@stu.hut.edu.cn (Z.L.); 17419002071@stu.hut.edu.cn (Q.M.); 2College of Electrical and Information Engineering, Hunan University, Changsha 410082, China; wangqii@hnu.edu.cn (Q.W.); szk@hnu.edu.cn (Z.S.)

**Keywords:** carbon emissions, green driving, green eco, reinforcement learning, policy gradient, platoon

## Abstract

Reinforcement learning (RL) methods for energy saving and greening have recently appeared in the field of autonomous driving. In inter-vehicle communication (IVC), a feasible and increasingly popular research direction of RL is to obtain the optimal action decision of agents in a special environment. This paper presents the application of reinforcement learning in the vehicle communication simulation framework (Veins). In this research, we explore the application of reinforcement learning algorithms in a green cooperative adaptive cruise control (CACC) platoon. Our aim is to train member vehicles to react appropriately in the event of a severe collision involving the leading vehicle. We seek to reduce collision damage and optimize energy consumption by encouraging behavior that conforms to the platoon’s environmentally friendly aim. Our study provides insight into the potential benefits of using reinforcement learning algorithms to improve the safety and efficiency of CACC platoons while promoting sustainable transportation. The policy gradient algorithm used in this paper has good convergence in the calculation of the minimum energy consumption problem and the optimal solution of vehicle behavior. In terms of energy consumption metrics, the policy gradient algorithm is used first in the IVC field for training the proposed platoon problem. It is a feasible training decision-planning algorithm for solving the minimization of energy consumption caused by decision making in platoon avoidance behavior.

## 1. Introduction

The aim of green autonomous driving is to enable a vehicle to navigate and make decisions based on its surroundings without human intervention while adhering to environmental protection standards. A critical component of achieving this goal is developing driving strategies that can automatically output control signals such as steering, throttle, and brake in response to the observed environment. 

The behavioral decision-making methods of automatic driving are divided into traditional methods and reinforcement learning methods. Traditionally, the decision control software system of an automatic driving system includes environmental prediction, behavior decision, action planning, path planning, and other functional modules. Traditional methods are often rule-based state control algorithms, including fuzzy logic [1], PID Bayesian control [2], and so on. Although these estimation algorithms are very accurate, such as Kalman filter [3], Kalman estimating IMU [4,5], Kalman estimating GNSS [5,6], and YOLO for RGB image detection [7], this paper focuses more on behavioral decision algorithms. For traditional behavioral decision making, behavioral decision algorithms have the advantages of easy construction and adjustment, good real-time performance, simple application, etc. However, because it is difficult to adapt to all situations, they need to make targeted adjustments and their behavioral rule base can easily overlap and fail, and it is difficult for a finite state machine to cover all the conditions that the vehicle may encounter, resulting in decision making errors. For behavioral decision making based on reinforcement learning, the influence of environmental uncertainties can be reduced by simulating and learning various unexpected situations due to the strong computing power. The traditional approach is to utilize multiple sensors, such as cameras [8], radar [9,10], and lidar [11], to map visual inputs directly to action outputs [12,13]. However, traditional methods for developing driving strategies, such as utilizing multiple sensors, can generate excessive heat and consume a significant amount of energy, which is not conducive to the green and low-carbon aims of autonomous driving. Furthermore, these methods can be cumbersome and require significant resources to develop and implement [14]. In contrast, reinforcement learning offers a promising alternative to traditional methods for developing driving strategies. Unlike these traditional methods, which rely on human supervision and can be cumbersome and resource-intensive, reinforcement learning is achieved through an iterative trial-and-error approach that does not require explicit human supervision. This technique is well suited to action planning and has shown promise in developing effective driving strategies [15]. On the other hand, little research has been conducted on whether autonomous driving is more energy efficient than manual driving, particularly with regard to obstacle avoidance strategies [16].

Veins [17] is an open-source framework that enables the simulation of wireless communication in mobile in-vehicle environments. It is a useful tool for studying topics such as autonomous vehicle driving, formation driving, path planning, and coordination in signalized areas within connected vehicle environments. The framework’s underlying structure can be used directly, allowing researchers to avoid wasting time and effort on non-research elements that can still significantly impact simulation results [18]. On the other hand, by exporting Veins simulations as OpenAI Gyms, Veins-Gym enables the use of reinforcement learning algorithms to address problems in the domain of vehicular ad hoc networks (VANETs) [19].

The connection established by Max Schettler between Veins tools and reinforcement learning by coupling Veins with OpenAI Gym provides a bridge for researchers who possess expertise in either field to leverage their knowledge in the other. This interface enables VANET researchers to access a compatible platform based on the generic RL framework [20]. Researchers can concentrate on studying algorithms and communication within the framework without the need to develop complicated interfaces for both Veins and reinforcement learning from scratch [21]. For platoon control, vehicle speed and acceleration are important state inputs, but these states cannot be obtained directly. Through sensor data such as GNSS, IMU, and camera, many scholars design robust estimation methods to obtain the states indirectly. However, in the Veins-Gym platform of this study, researchers do not need to pay much attention to environment building and state acquisition beyond reinforcement learning algorithms.

The field of reinforcement learning for energy conservation, green economy, and reducing CO_2_ emissions is extensive and encompasses a wide range of studies [22]. For example, in the optimal regulation of microgrids, carbon emissions allowances take into account volume prediction [23], electrical energy consumption management of household appliances [24], hybrid clouds harnessing renewable energy, task scheduling [25], and so on, which all concern green and low-carbon environmental protection. However, despite the significant attention given to reinforcement learning applications for energy conservation and carbon emission reduction, limited studies have addressed the issues related to autonomous driving, connected vehicle communication, and energy consumption. These issues include carbon emissions from the decision-making behavior of autonomous vehicles, energy consumption during connected vehicle communication, etc., and how to reduce greenhouse gas emissions from vehicles. Max Schettler’s development of Veins-Gym has made it easy to perform data statistics and analysis, offering an excellent simulation and development tool for investigating fleet behavior decision making [26,27], particularly the obstacle avoidance problem on which this paper concentrates [28,29].

This paper focuses on examining how reinforcement learning algorithms can be utilized to train the behavior of member vehicles in a CACC platoon [30,31] consisting of vehicles with varying parameters in the event of a serious collision involving the leading vehicle [32]. Additionally, the paper aims to determine the most energy-efficient and eco-friendly solution that consumes the least amount of energy while also fulfilling the requirements of avoiding collisions or minimizing collision damage. The conventional approach relies on using sensors that operate independently, without any communication, to assess the surroundings and determine the appropriate course of action. 

The main contributions of this paper are as follows:A hypothetical situation is created to depict a scenario where a line of vehicles is present on a two-lane highway. When there are no other vehicles present, the front vehicle of the platoon suffers a serious traffic accident.To prevent further damage, reinforcement learning (policy gradient algorithm) is used to obtain the most efficient strategy to be adopted by the member vehicles to reduce the impact of the collision on the team.While solving the collision avoidance problem, the reinforcement learning algorithm also examines the damage caused by the vehicle behavior and computes the strategy that minimizes the damage.In order to break the limitations of traditional algorithms, reinforcement learning algorithms (policy gradients) are applied to the behavioral decision of the fleet, which is a leap forward and a hot spot for future research.

## 2. Platoon Algorithms with RL

This section focuses on several aspects, including inferring the formula of the policy gradient (PG) algorithm for reinforcement learning, modeling the Veins simulation model, and assuming a universal and typical simulation scenario [33].

### 2.1. The CACC Car-Following Model

$x_{i}$ is the displacement of the following vehicle, $v_{i}$ is the speed of the following vehicle, e is the error between the actual distance and the desired distance, T is the minimum safe headway time distance, i−1 is the front car, and $v_{kprev}$ is the speed of the vehicle in the previous moment.
(1)$$\begin{array}{c} v_{i}=v_{kprev}+k_{pe}+k_{d}\dot{e}\\ e=x_{i-1}-x_{i}-Tv_{i} \end{array}$$

### 2.2. Proposed Car Dynamics Cost Model

This paper assumes that the nature of the collision between vehicles is inelastic:(2)$$\{\begin{array}{c} m_{2}v_{0}=m_{1}v_{1}+m_{2}v_{2}\\ \frac{1}{2}m_{2}v_{0}^{2}=\frac{1}{2}m_{1}v_{1}^{2}+\frac{1}{2}m_{2}v_{2}^{2}+E \end{array}$$

For individual vehicle loss functions within a fleet:(3)$$J_{i}=E_{i}=\frac{1}{2}m_{2}\left(v_{0}^{2}-v_{2}^{2}\right)-\frac{1}{2}m_{1}v_{1}^{2}\;$$

In contrast to the actual scenario, Wentao Chen posits that the nature of vehicle collisions is completely inelastic. However, the analysis and reconstruction of heavy goods vehicle traffic accidents categorizes the nature of vehicle collisions based on three distinct properties associated with three different vehicle speeds. This approach aligns more accurately with real-world situations. In this paper, the relationship between mass and collision loss is proposed as:(4)$$\frac{v_{2}}{v_{1}}=\frac{1}{2}\frac{1}{1+e^{\ln\frac{m_{1}}{m_{2}}}}=\frac{1}{2}\frac{m_{2}}{m_{1}+m_{2}}$$

To facilitate the calculation, we specify:

When $m_{1}<m_{2}$, it is elastic collision;
$v_{2}/v_{1}=1/2$.When $m_{1}=m_{2}$, it is inelastic collision; $v_{2}/v_{1}=1/4$.When $m_{1}>m_{2}$, it is completely inelastic collision; $v_{2}/v_{1}=0$.

The platoon cost function is defined as:(5)$$J=\overset{N}{\underset{i=1}{\sum }}J_{i}$$

### 2.3. Markov Decision Process (MDP)

For a complete sequence of state behavior trajectories $\tau =\left(s_{0},a_{0},\ldots ,s_{T-1},a_{T-1},s_{T}\right)$, i.e., the process of obtaining the next state $s_{i+1}$ after obtaining behavior $a_{i}$ at state $s_{i}$, there are single-step reward functions $R\left(s_{t},a_{t}\right)$ and total reward functions $R\left(\tau \right)=\sum_{t=0}^{T-1}\gamma^{i}R\left(s_{t},a_{t}\right)$ for obtaining the maximum expected reward, where $\pi_{\theta }$ is the parameterization strategy of the neural network composition. Under the strategy $\pi_{\theta }$, the expected value *τ* is used for the trajectory, so the problem can be transformed into finding the optimal parameter θ.

The gradient of the desired reward can be obtained from the gradient descent $\nabla_{\theta }E_{\pi \theta }R\left(\tau \right)$, and the parameters are updated by the hyperparametric learning rate α.
(6)$$\theta \leftarrow \theta +\alpha \nabla_{\theta }E_{\pi \theta }R\left(\tau \right)\;$$

Let P(τ|θ) be the probability of the trajectory τ under the strategy $\pi_{\theta }$. Then, the gradient can be calculated:(7)$$\nabla_{\theta }E_{\pi \theta }R\left(\tau \right)=\nabla_{\theta }\underset{\tau }{\sum }P\left(\tau |\theta \right)R\left(\tau \right)\;=\underset{\tau }{\sum }\nabla_{\theta }P\left(\tau |\theta \right)R\left(\tau \right)\;=\underset{\tau }{\sum }\frac{P\left(\tau |\theta \right)}{P\left(\tau |\theta \right)}\nabla_{\theta }P\left(\tau |\theta \right)R\left(\tau \right)\;=\underset{\tau }{\sum }P\left(\tau |\theta \right)\nabla_{\theta }\log P\left(\tau |\theta \right)R\left(\tau \right)\;=E_{\pi \theta \left(\nabla_{\theta }\log P\left(\tau |\theta \right)R\left(\tau \right)\right)}$$

Therefore, the probability of the trajectory τ can be calculated:(8)$$P\left(\tau |\theta \right)=p\left(s_{0}\right)\overset{T-1}{\underset{t=0}{\prod }}p\left(s_{t+1}|s_{t},a_{t}\right)\pi_{\theta \left(a_{t}|s_{t}\right)}$$
where $p(s_{t+1}|s_{t},a_{t})$ is the probability of transitioning to state $s_{t+1}$ after taking behavior $a_{t}$ at the moment of state $s_{t}$.

### 2.4. Policy Gradient Algorithm

Consider the optimization model from the gradient of the objective function:$$\nabla J_{\theta }\left(\theta \right)=\int \nabla_{\pi \theta }\left(\tau \right)r\left(\tau \right)dr=E_{\tau \sim \pi \theta \left(\tau \right)}\nabla_{\theta }\log\pi_{\theta }\left(\tau \right)r\left(\tau \right))$$
to obtain the gradient:(9)$$E_{\tau \sim \pi \theta \left(\tau \right)}\left(\underset{t}{\sum }\nabla_{\theta }\log\pi_{\theta }\left(a_{t}|s_{t}\right)\right)\left(\underset{t}{\sum }r\left(s_{t},a_{t}\right)\right)$$

Strategy function:

When designing policy functions, it is important to address the discrete state space and the continuous state space separately due to their substantial differences in the quantity and definition of states.

### 2.5. Gauss Policy Function

The Gauss strategy function for the continuous behavior space is generated from a Gaussian distribution with a fractional function:(10)$$\nabla_{\theta }\log_{\pi \theta }\left(s,a\right)=\frac{\left(a-\phi {\left(s\right)}^{T_{\theta }}\right)\phi \left(s\right)}{\sigma^{2}}$$

### 2.6. Softmax Policy Function

Softmax for discrete spaces:(11)$$\pi_{\theta }\left(s,a\right)=\frac{e^{\phi {\left(s,a\right)}^{T_{\theta }}}}{\sum_{b}e^{\phi {\left(s,a\right)}^{T_{\theta }}}}$$

The odds of a behavior occurring are weighed using a linear combination of the features ϕ(s,a) describing the state and the behavior with the parameter θ. The corresponding score function is its derivative:(12)$$\nabla_{\theta }\log_{\pi \theta }\left(s,a\right)=\phi \left(s,a\right)-E_{\pi \theta }\left[\phi \left(s,\cdot \right)\right]$$

## 3. Proposed Model

### 3.1. Vehicle Dynamics and Network Parameters

Consider a smooth and empty two-lane, one-way straight highway with a row of N convoys with the following parameters.

The fleet uses the CACC follow-the-leader model such that the fleet member i∈N can maintain a relative speed of 0. 

Let the lead vehicle in the convoy with marker i=0 at the moment t=0 have a larger vehicle collision, so that:(13)$$v_{0}=\{\begin{array}{cc} 120, & t<0\\ 110-8t, & t\geq 0\\ 0, & t\geq 13.75 \end{array}$$

Vehicle 1, when vehicle 0 experiences the accident, brakes in response when the relative distance is reduced to 6.8 m; when the vehicle speed is reduced to 0, the relative speed is reduced to 57.16 m, which is greater than the head time distance of 1.5 s, that is, 50 m, resulting in vehicle 1 causing a rear-end accident and a chain of further rear-end accidents from the cars behind it. 

Consider the workshop communication using protocol IEEE802.11p for short-distance communication. To verify and confirm the network adjacency, hello packets are sent periodically by the vehicle. To ensure the timeliness of the communication, the transmission protocol uses the faster UDP and the packet interval is set to 0.2 s. Learning rate α and exploration rate ε are the reinforcement learning training parameters. The detailed simulation parameters are shown in Table 1.

Additionally, considering the scenario, the mass of the motor vehicle does not satisfy $m_{i}=m_{j}$ and the deceleration does not satisfy $a_{i}=a_{j}$; therefore, in scenarios where traffic accidents are caused by the leading vehicle, braking may not be the optimal approach. The growing volume of road traffic necessitates further reduction in driving distances, which in turn calls for increased attention to the safety of autonomous vehicles. It is essential to focus on developing more intelligent approaches for path planning and safe decision making. Policy gradient algorithm for platoon is as follows (Algorithm 1).

**Algorithm 1** Policy gradient for platoonInput: a differentiable policy parameterization
π(a|s,θ)
Algorithm parameter: step size
a>0
Initialize policy parameter
$\theta \in R^{d^{\prime }}$, environment and state
$S_{0}$

1 Loop for each episode:2 Generate an episode
$S_{0},A_{0},R_{1},\ldots ,S_{T-1},A_{T-1},R_{T}$, following
π(·|·,0)
3 Loop for each step of the episode t=0,…, *T*−1:
4 S← return from step t ($S_{t}$)5 

$R\leftarrow R+\gamma^{i}R(s_{t},a_{t})$

6 

$\theta \leftarrow \theta +\alpha \gamma^{t}\nabla {\overset{-}{R}}_{\theta }$

7 If episode is complete:8 Break9 Train and learn for agent:10Return *θ*


### 3.2. Strategic Gradient Decision-Making Behavior

#### 3.2.1. State Space and Action Space

In certain public platforms, such as Gym, the state space for most domains is readily accessible, allowing scholars to compare the performance and convergence speed of various algorithms. However, in real-world projects, state space design work must be carried out independently. Based on the author’s personal experience, adding new state information can significantly improve performance, more so than other aspects (such as tuning), which is very cost effective, so the optimization of the state space is almost always carried out in the project.

#### 3.2.2. Mission Analysis

The following situation was established: the foremost vehicle of a convoy on a highway becomes engaged in a significant collision, resulting in a severe reduction in speed. At this point, the convoy members are compelled to make crucial choices to evade the collision or decrease the harm caused by an unavoidable collision.

#### 3.2.3. Observation Spatial Information Filtering

The information in the environment is passed to the intelligence (agent) for generating the reward function (reward). In this paper, we consider that for the ith member car, the key point is the relative position, the relative velocity of the previous car i-1 and i-2. Moreover, if the car decides to change lanes, the environmental feedback data of the next state, which include the relative position and velocity with the preceding car after changing lanes, are analyzed. The state space is defined as [x1, x2, x3, v1, v2, v3], where x and v denote the distance and speed, respectively, and the labels 1, 2, and 3 denote the physical quantities with the car in front, the car behind, and the car in front of the next lane, respectively. The behavior space comprises [lane change, deceleration], and their elements are Boolean values.

#### 3.2.4. Reward Function Settings

The fleet collision loss function is utilized to consider the environment space key points. The reward in the next moment within the current lane is dependent on the state space, where any behavior $a_{i}$ must result in a new change in the environment $s_{i}$. To ensure the safety of all vehicles and that each vehicle remains within the safety threshold, the reward function must be designed accordingly. Additionally, the reward function takes into account the fuel consumption that results from the vehicle’s braking and lane-changing behaviors.

## 4. Analysis of Simulation Results

The convergent process of each loss function value in the whole training process is compared under different learning and exploration rates. The energy loss function and the collision loss function are slightly different. The two are not strongly correlated. When the energy loss is low, the collision loss may be larger; sometimes, the opposite result is produced. The selection of super parameters is extremely important. The training system is chaotic, and any small change in the parameter value leads to great changes in the convergence characteristics, even in the non-convergence situation. The value of the learning rate parameter used for the constraint training convergence curve is set to [0.001, 0.1]. The exploration rate parameters used to explore more decisions have values of [0.001, 0.2].

Considering the complexity of the scenario, it is reasonable to assume that there is no need to dynamically adjust the learning rate. Figure 1, Figure 2, Figure 3 and Figure 4 show the loss calculated by the strategy gradient algorithm according to the training time under the condition of different learning rates and exploration rates in the target scenario. In order to obtain faster convergence time and reduce training time in the whole training process, the value of the learning rate should be larger. A low learning rate leads to a difficult training process and long convergence time. Too high a learning rate causes a serious loss value jump, makes it difficult to obtain the overall downward trend, and even causes the problem of non-convergence. In order to obtain more strategy choices during the initial training, the value of the exploration rate needs to be set high. Too low an exploration rate makes it difficult to jump out of the current decision, that is, the loss gradient drops to a minimum point that is not relatively good and the existing decision remains unchanged. Too large an exploration rate causes the agent to hesitate, try repeatedly among many possible choices, and fail to make the optimal decision. Based on the training results, the optimal learning rate, α = 0.05, and the optimal exploration rate, ε = 0.1, are obtained.

In the simulation scenarios and algorithm models of Veins-Gym, the vehicle under test continuously tests and trains the deep neural network. Sensor-based vehicle distance decision algorithms need no training. The decision of CACC is determined by the headway, min-Gap (the distance between the front bumper and the rear bumper of the vehicle in the queue), and the reaction time τ. The optimal headway = 1.5 s, min-Gap = 2.5 m, and τ = 1 s are adopted. Comparing the traditional algorithm with the proposed algorithm, the following is obtained.

In one thousand training sessions, a target vehicle exhibits a pattern of both overall convergence and local oscillation in the collision loss generated by the fleet and the reward value it receives. The proposed method differs from the traditional approach in that lane changes occur between vehicles instead of all vehicles changing lanes. The loss convergence process of the agent, which is highly correlated with the reward function, can be evaluated using multiple metrics. When evaluating a single intelligent entity, the most crucial factor to consider is the trend of collision loss and reward value changes.

There is no strong correlation between energy loss and collision loss, which is particularly obvious in the traditional algorithm; that is, all CACCs of the team members may adopt the same lane change strategy at the same time, which makes the decision without safety significance and increases energy consumption. According to the Figure 5, the policy gradient algorithm-based reinforcement learning method outperforms conventional methods in anticipating autonomous driving situations, devising optimal routes, and issuing timely warnings. This superiority is evident in both subjective impressions and objective assessments.

### Analysis of Three-Lane Scenario Simulation Results

In the three-lane experiment, the parameters and the design of the model were different compared to those for the two-lane experiment. According to Figure 6,the distance and steering model was introduced here, which is more in line with the actual situation. The expansion region is defined as follows: the centroid of the top view of the car is the center of the expansion rectangle; the side length is the rectangular edge, which is determined by the car edge and the speed of the two vehicles. The minimum distance between the rectangles corresponds to the safety car distance of the two vehicles. When the following vehicle’s speed is greater than that of the leading vehicle and the lane changes, the velocity vector at the minimum distance point is used to predict the collision and decide the change in the reward function. When the speed of the following vehicle is close to or less than the speed of the leading vehicle, the velocity vector has little significance. It is used to share the speed of the forward direction and increase the distance between the two vehicles.

The operation loss parameter of the reward function is set to a larger value, that is, the reward value is more affected by the operation loss. The vehicle trajectories over a period of time are shown in the Figure 7 and Figure 8 below. The agent should pay as little attention as possible to the influence of the distance change outside the expansion interval on the security. Agents are trained several times, and the three agents with the maximum reward value are taken as the training results. There are some commonalities among these three results. The first and second cars are always in different lanes, while the third car has no obvious correlation with the position of the car in front. This is because the third car is furthest away from the accident vehicle and therefore has more maneuvering time and distance.

Examining the relationship between the sequence of fleet members and important physical quantities over time is of great interest. Even though vehicles make many steering movements during high-speed driving, the results of experiments conducted in a reinforcement learning agent simulation environment are worth investigating. Even if the directional change resulting from this movement is very small, typically less than 10 degrees, it can have catastrophic consequences. For instance, in rainy or snowy weather, the friction between the tire and the ground in the vertical direction to the direction of travel can cause rolling friction which transforms into sliding friction. Once this occurs, the vehicle’s state becomes uncontrollable, leading to serious traffic accidents. In this three-lane experiment, the behavior of each member is intrinsically linked to the entire system. Due to communication delay and uncontrollable hardware control random delay, the behavior of each vehicle has a reasonable lag and delay in response to the previous vehicle’s actions.

The function that describes the change in the minimum distance between a vehicle and its leading vehicle as a function of time is always monotonically decreasing, but for vehicles with different indices, the curves exhibit reasonable differences. In general, vehicles with higher indices exhibit greater negative acceleration, i.e., they tend to maintain a greater distance from their leading vehicle. Acceleration, as the second derivative of distance, must have a sufficiently large integral to increase the distance between vehicles; therefore, the acceleration of rear vehicles must be larger and initiated earlier to obtain a larger reward for the agent. To increase the distance between vehicles under loose conditions, vehicles can gain greater distance by making free turns and agents will do whatever it takes, including changing lanes continuously, to achieve this goal. When the vehicle that continuously changes lanes reaches the front of the road and no longer detects a leading vehicle, it is not motivated to change lanes or decelerate. Although the vehicle in this situation should accelerate and overtake surrounding vehicles to leave the accident area as quickly as possible, the simulation environment does not have parameters or behaviors that are less related to obstacle avoidance, minimal energy loss, and minimal safety cost. Therefore, the acceleration of the vehicle is always non-positive, sometimes negative, and sometimes zero.

## 5. Conclusions

In this paper, we propose a policy gradient algorithm for computing the minimum energy consumption for autonomous driving decisions in the context of a green economy. The approach utilizes the Veins-Gym reinforcement learning and the Telematics training platform. Two experiments were designed to train the proposed algorithm for obstacle avoidance strategies in two-lane and three-lane highway situations when the team’s lead vehicle is involved in a major traffic accident. By training the optimal decision, the vehicles perform better in terms of obstacle avoidance and energy loss performance and achieve minimized cost loss. Due to different parameter settings, the vehicles’ obstacle avoidance strategy can be switched between conservative and aggressive operations. In the future deployment of a large number of self-driving vehicles, reinforcement learning has great promise for some decisions, such as environmental prediction and behavioral decisions. 

Future research should address the following: This study used short-range communication networks only for small-scale vehicle communication, and when more vehicles and more network types (such as base stations) are added, how reinforcement learning can cope with them should be specifically analyzed.When the road environment is more complex, reinforcement learning intelligences should be multi-agents and consideration should be given to whether to use distributed or centralized agents.The crash environment in this study is only one of many small probability situations; a more generalized decision algorithm should be sought to minimize loss.There is endogeneity in the road passage, traffic calming methods, and crash behavior decisions, and their correlation should be studied.

## Figures and Tables

**Figure 1 sensors-23-04176-f001:**
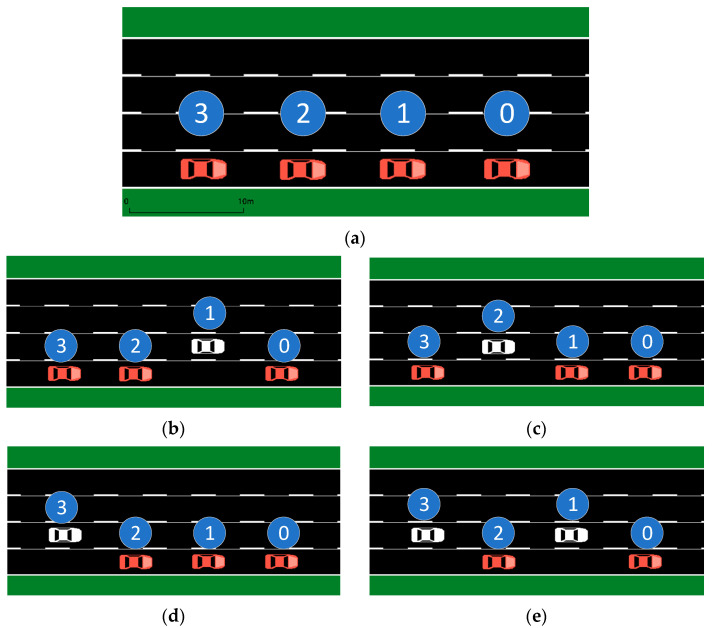
Decision making: slowing down and changing lanes. (**a**) No action; (**b**) the 1st member car changing lanes; (**c**) the 2nd member car changing lanes; (**d**) the 3rd member car changing lanes; (**e**) the 1st and 3rd member cars changing lanes.

**Figure 2 sensors-23-04176-f002:**
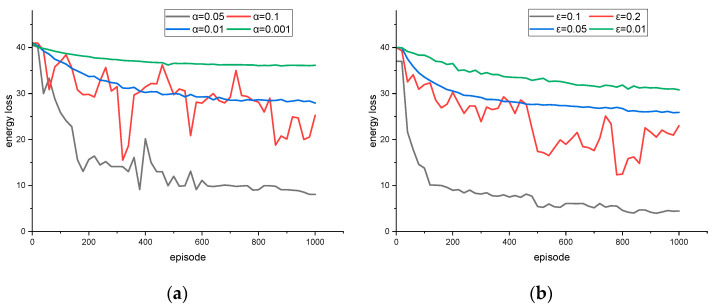
In order to obtain the optimal energy loss strategy, different hyperparameters were used for training energy loss with (**a**) different α; (**b**) different ε.

**Figure 3 sensors-23-04176-f003:**
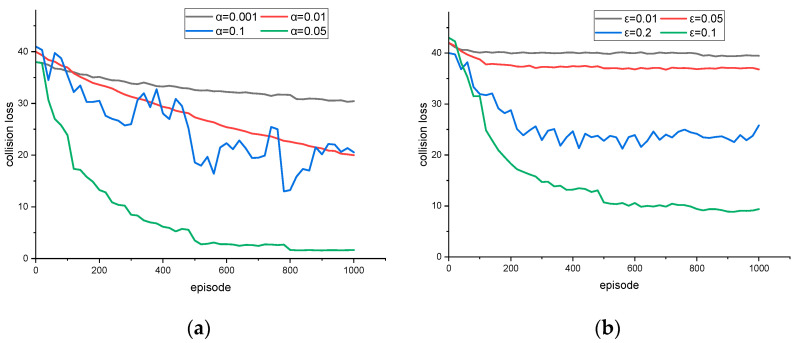
In order to obtain the optimal collision loss strategy that is different from energy loss, different hyperparameters were used for training collision loss with (**a**) different α; (**b**) different ε.

**Figure 4 sensors-23-04176-f004:**
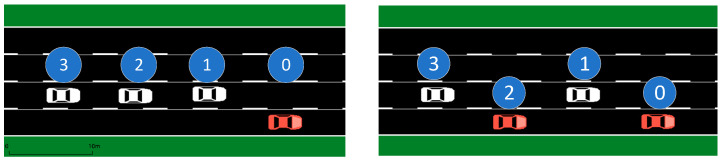
Result of traditional and proposed method.

**Figure 5 sensors-23-04176-f005:**
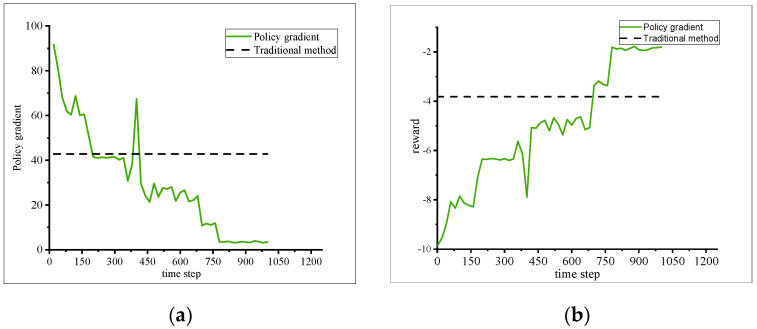
PG algorithm and traditional algorithm cost function graph. (**a**) Collision loss convergence; (**b**) reward convergence.

**Figure 6 sensors-23-04176-f006:**
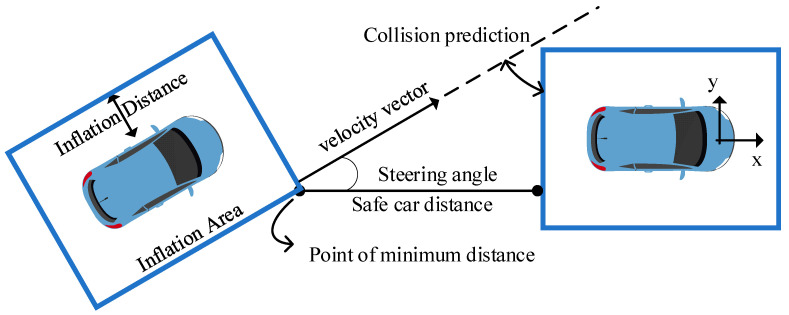
Distance and steering model.

**Figure 7 sensors-23-04176-f007:**
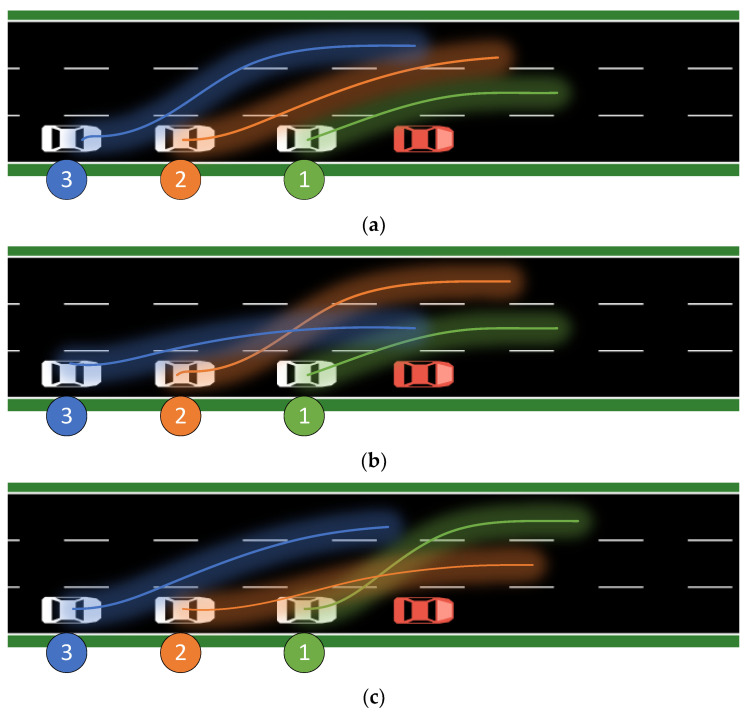
Results of agent training with minimum operational loss priority. The trajectories of (**a**) car 1 changing to lane 2; car 2, 3 to 3; (**b**) car 1, 3 changing to lane 2; car 2 to 3; (**c**) car 1 changing to lane 2; car 2, 3 to 3.

**Figure 8 sensors-23-04176-f008:**
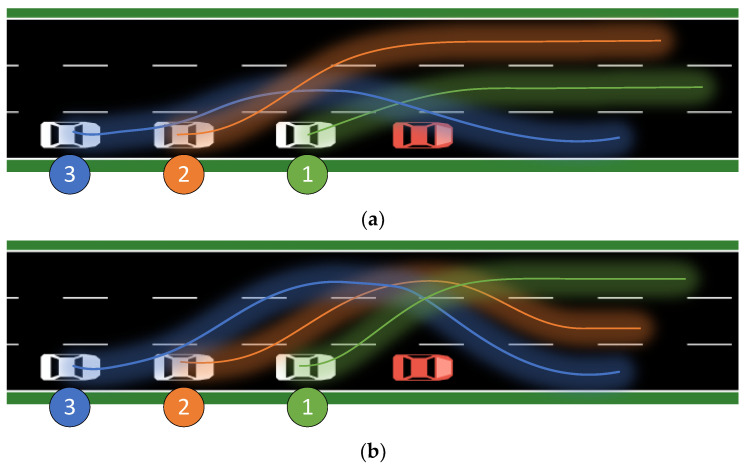
Results of agent training with large distance loss priority. Number of lane changes for vehicle 1, 2, 3: (**a**) 1, 2, 2; (**b**) 2, 3, 4; (**c**) 1, 2, 4.

**Table 1 sensors-23-04176-t001:** Assumed speed and routing protocol.

Parameter	Value
Car speed	120 Km/h
Headway	1.5 s
Response time	1 s
Deceleration	8 m/s^2^
Routing packet size	512 Bytes
Simulation distance	1000 m
Maximum rate	10 MB/s
Number of nodes	4
Communication distance	250 m
Hello packet interval	Ls
Transfer Protocol	UDP
Packet interval	0.1 s
MAC layer protocol	802.11
Channel transmission rate	3 Mbps
Learning rate α	0.001, 0.01, 0.05, 0.1
Exploration rate ε	0.01, 0.05, 0.1, 0.2

## Data Availability

Not applicable.

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
