# Peer review of "Reinforcement Learning-Based Approach for Minimizing Energy Loss of Driving Platoon Decisions"

_sensors, 2023, doi:10.3390/s23084176_

Round 1

Reviewer 1 Report

the current version cannot be accepted

Author Response

Please see the attachment. The relevant parts of the revision manuscript have been color coded.

Reviewer 2 Report

The paper is very difficult to follow and read.

English writing has to be fixed, and very long statements should be avoided.

The introduction contains all related work studies; however, a separate section should be introduced for the related work that compares previous studies and the newly proposed model to clarify its importance. More previous studies can be investigated in this field.

The title of section 2 should be more expressive.

Section 3 should be divided into two sections. The first introduces the solution, while the second analyzes its performance.

The considered metrics and investigated parameters have to be clearly explained.

The finding and results should be highlighted.

A comparative study with previous research in this field is required.

The conclusion is very long, and several details can be omitted.

Some future enhancement directions could be declared for this study.

Author Response

(The authors gave the same response as above.)

Reviewer 3 Report

This paper proposes a method for driving platoon decisions based on RL. My comments are as follows:

1: The idea is interesting and the method has some potential. The authors also conducted extensive simulation experiments to demonstrate their work.

2: For the model of Platoon with RL, vehicle velocity and acceleration are important state input. However, these states could not be obtained directly. Many scholars choose to design robust estimation methods to obtain it indirectly. With new sensors configuration for autonomous vehicles, such as GNSS, IMU, and cameras, the vehicle states could be estimated precisely. Thus, some related work should be included: autonomous vehicle kinematics and dynamics synthesis for sideslip angle estimation based on consensus kalman filter, estimation on imu yaw misalignment by fusing information of automotive onboard sensors, automated vehicle sideslip angle estimation considering signal measurement characteristic, yolov5-tassel: detecting tassels in rgb uav imagery with improved yolov5 based on transfer learning, imu-based automated vehicle body sideslip angle and attitude estimation aided by gnss using parallel adaptive kalman filters, improved vehicle localization using on-board sensors and vehicle lateral velocity, automated driving systems data acquisition and processing platform.

3: It would be better to highlight your contribution in the introduction.

4. Please list the environment and parameter setting for RL in detail.

5. Future work should be included at the end of the conclusions.

Author Response

(The authors gave the same response as above.)

Round 2

Reviewer 2 Report

The paper has been extensively improved.

All of my comments have been addressed.

I think this version of the paper can be accepted